# *CaSPDS*, a Spermidine Synthase Gene from Pepper (*Capsicum annuum* L.), Plays an Important Role in Response to Cold Stress

**DOI:** 10.3390/ijms24055013

**Published:** 2023-03-06

**Authors:** Jianwei Zhang, Minghui Xie, Guofeng Yu, Dong Wang, Zeping Xu, Le Liang, Jiachang Xiao, Yongdong Xie, Yi Tang, Guochao Sun, Bo Sun, Zhi Huang, Yunsong Lai, Huanxiu Li

**Affiliations:** 1College of Horticulture, Sichuan Agricultural University, Chengdu 611130, China; 2Institute for Processing and Storage of Agricultural Products, Chengdu Academy of Agricultural and Forest Sciences, Chengdu 611130, China; 3Institute of Pomology and Olericulture, Sichuan Agricultural University, Chengdu 611130, China

**Keywords:** spermidine synthase, virus-induced gene silencing, overexpression, cold stress, pepper

## Abstract

Spermidine synthase (SPDS) is a key enzyme in the polyamine anabolic pathway. *SPDS* genes help regulate plant response to environmental stresses, but their roles in pepper remain unclear. In this study, we identified and cloned a SPDS gene from pepper (*Capsicum annuum* L.), named *CaSPDS* (LOC107847831). Bioinformatics analysis indicated that *CaSPDS* contains two highly conserved domains: an SPDS tetramerisation domain and a spermine/SPDS domain. Quantitative reverse-transcription polymerase chain reaction results showed that *CaSPDS* was highly expressed in the stems, flowers, and mature fruits of pepper and was rapidly induced by cold stress. The function of *CaSPDS* in cold stress response was studied by silencing and overexpressing it in pepper and *Arabidopsis*, respectively. Cold injury was more serious and reactive oxygen species levels were greater in the *CaSPDS*-silenced seedlings than in the wild-type (WT) seedlings after cold treatment. Compared with the WT plants, the *CaSPDS*-overexpression *Arabidopsis* plants were more tolerant to cold stress and showed higher antioxidant enzyme activities, spermidine content, and cold-responsive gene (*AtCOR15A*, *AtRD29A*, *AtCOR47*, and *AtKIN1*) expression. These results indicate that *CaSPDS* plays important roles in cold stress response and is valuable in molecular breeding to enhance the cold tolerance of pepper.

## 1. Introduction

Cold stress, which includes chilling (0–15 °C) and freezing (<0 °C), is an adverse environment condition that greatly affects plant growth, development, and survival and also constrains the geographical distribution of plants [1,2]. Generally, exposing temperate plants to low temperatures (>0 °C) for a period could increase their freezing tolerance. However, several plant species, such as tomato, pepper, tobacco, and rice, which are from the tropics and subtropics, are sensitive to cold stress [2,3]. To minimise cold injury and improve survival rate, plants have evolved systemic defence mechanisms, including changes in plasma membrane components, synthesis of osmotic substances, and expression of cold-responsive (COR) genes [4,5,6].

Polyamines (PAs) are low-molecular-weight aliphatic amines that exist widely in prokaryotic and eukaryotic cells. Common PAs in plants include putrescine (Put), spermidine (Spd), and spermine (Spm), which play important roles in many basic physiological processes and stress responses [7,8,9,10,11]. The homeostasis of PAs in plants is affected by de novo synthesis and catabolism. The biosynthesis of PAs is catalysed by several enzymes, the most important of which include arginine decarboxylase (ADC), ornithine decarboxylase (ODC), S-adenosylmethionine decarboxylase (SAMDC), spermidine synthase (SPDS), and spermine synthase (SPMS) [12]. ADC and ODC catalyse the synthesis of Put from L-arginine and L-ornithine, respectively. SAMDC catalyses the decarboxylation of S-adenosylmethionine (SAM) to provide one and two aminopropyl groups for Put and Spd, respectively, and then SPDS and SPMS catalyse the above reaction to produce Spd and Spm, respectively [13,14]. Meanwhile, the catabolism of PAs is only mediated by polyamine oxidase and diamine oxidase [15].

To date, several genes related to PA biosynthesis have been cloned and characterised from various plant species [11,16,17]. *SPDS* genes help regulate the flowering and fruit quality of plants [18,19,20]. Furthermore, SPDS has been associated with responses to biotic and abiotic stresses. Qiu et al. [21] found that exogenous Spd can improve the bacterial wilt resistance of eggplant and that *SmSPDS* silencing through virus-induced gene silencing (VIGS) increases plant susceptibility to diseases. Neily et al. [22] found that overexpression of the apple *MdSPDS1* gene in tomato enhances plant tolerance to salt. In rice, *OsSPDS* participates in the response to chilling stress but not salt stress [23]. In addition, transgenic *Arabidopsis* and sweet potato expressing the *FSPD1* (*CfSPDS*) gene of *Cucurbita ficifolia* are more tolerant to abiotic stresses, including chilling temperature, drought, and salinity, than their wild types (WTs) [24,25]. Although the roles of SPDS have been studied in many species [26,27,28], its function remains unclear to date in pepper.

Pepper (*Capsicum annuum* L.), belonging to the Solanaceae family, is an important vegetable native to tropical America and an industrial raw material for spices, cosmetics, and medicine. China is becoming the largest producer and consumer of pepper worldwide. The cold stress in early spring affects the growth and development of pepper, especially at the reproductive stage. A previous multi-omics analysis showed that the expression of *SPDS* increased significantly under cold stress, suggesting that this gene regulates cold stress response [29]. Thus, the present study aimed to explore the biological function of *CaSPDS*, an *SPDS* gene isolated from *C. annuum*, in response to cold stress by silencing and overexpressing it in pepper and *Arabidopsis*, respectively. The isolated gene was subjected to bioinformatics and gene expression analyses. The results of this study contribute to our understanding of the roles of *CaSPDS* and provide a theoretical basis for improving the cold tolerance of pepper via breeding.

## 2. Results

### 2.1. Identification and Bioinformatics Analysis of CaSPDS in Pepper

*CaSPDS* (LOC107847831) was identified in the pepper genome through a BLAST sequence analysis the *Arabidopsis* (AT1G23820 and AT1G70310) and rice (LOC4342996) genes against the NCBI database. The ORF sequence length of *CaSPDS* was 1023 bp, encoding 341 amino acids, and the predicted protein weight was 37.59 kDa. The pI of *CaSPDS* was 5.1. The instability index of *CaSPDS* was greater than 40, suggesting that it is unstable (Figure 1A). *CaSPDS* contained 9 exons and 8 introns and located at 11th chromosomes (Figure 1B). The hydropathicity value of *CaSPDS* varied from −3.167 to 1.922 (Appendix A), and neither transmembrane domains nor signal peptides were detected (Appendix A). The phosphorylation sites were predicted as follows: 13 serine, 8 threonine, and 4 tyrosine (Appendix A). The secondary structure of CaSPDS was predicted to comprise 35.48% alpha helices, 16.42% extended strands, 7.92% beta turns, and 40.18% random coils (Figure 1C). Based on these special structures, the tertiary structure of CaSPDS is shown in Figure 1D.

### 2.2. Phylogenetic and Conserved Domain Analyses of SPDS in Different Species

An evolutionary tree was generated using MEGA-X software to understand the evolutionary relationship between CaSPDS and SPDS proteins from 15 other species. The CaSPDS protein was clustered together with proteins that were derived from Solanaceae plants, such as tomato, potato, petunia, common tobacco, and woodland tobacco. Interestingly, the SPDS proteins of *Solanaceae* plants are more closely related to those from maize and rice than to those of lettuce, gentian, and Brassicaceae plants (Figure 2A). Subsequently, a multiple sequence alignment was performed on SPDS proteins of *Solanaceae* and *Arabidopsis* (Figure 2B). All SPDS enzymes contain two highly conserved domains, SPDS tetramerisation domain and SPMS/SPDS domain, which are composed of 55 and 189 amino acids, respectively. The pepper CaSPDS protein shared 86.79–92.77% identity with SPDS from other *Solanaceae* plants and showed high levels (77.55% and 79.38%) of identity with *Arabidopsis* AtSPDS and AtSPDS2, respectively (Figure 2B).

### 2.3. Expression Patterns of CaSPDS with and without Cold Stress

To understand the potential functions of *CaSPDS*, we examined its transcript level in different tissues by using quantitative reverse-transcription polymerase chain reaction (qRT-PCR). Samples were collected from six-leaves stage (young root, YR; young stem, YS; young leaf, YL; and cotyledon, C), flowering stage (flower, F), and fruiting stage (young fruit, YF; mature fruit, MF; mature root, MR; mature stem, MS; and mature leaf, ML). As shown in Figure 3A, *CaSPDS* was highly expressed in MS, followed by MF, F, and YS. This gene had similar transcription levels in YR, YL, YF, MR, and ML. Notably, the expression of *CaSPDS* in C was very low. In addition, we measured *CaSPDS* expression in response to cold stress by qRT-PCR and available RNA-seq data [29]. *CaSPDS* expression significantly increased in the early phase of treatment, peaked at 6 h, and then decreased in the late phase of treatment compared with that in the control (0 h) (Figure 3B). The transcriptome data showed the same tendency of initially increasing and then decreasing (Figure 3C). 

### 2.4. Subcellular Localisation and Prokaryotic Expression of CaSPDS

pBWA(V)HS-*CaSPDS*-GLosgfp was constructed and injected into tobacco leaves for transient expression to detect the subcellular localisation of *CaSPDS* (Figure 4A). Free-GFP was distributed in all parts of the cell, whereas *CaSPDS*-GFP only fluoresced in the nucleus. This result confirms that CaSPDS is a nucleus-localised protein (Figure 4B). p-COLD-*CaSPDS* encoding 6 × His-tagged CaSPDS fusion protein was constructed and transfected into *E. coli* BL21 (DE3) cells. As shown in Appendix A, 0.4 mM IPTG and 24 h induction were the most suitable expression conditions. The supernatant (Lan 2) contained the band of the target protein, indicating that CaSPDS was a soluble protein. In addition, the single purified protein (Lan 3) was also observed in 37.6 kDA band of the marker (Appendix A).

### 2.5. Influence of CaSPDS Silencing on Cold Stress Tolerance in Pepper

The expression of *CaSPDS* was highly induced by cold stress (Figure 3B,C), implying that this gene is involved in stress response. We therefore analysed the regulatory effect of *CaSPDS* on the cold stress tolerance of pepper through VIGS. At 4 weeks after transient transformation, leaf albinism was observed in the TRV:*PDS* (positive control) plants but not in the TRV:00 (negative control) and TRV:*CaSPDS* plants (Appendix A), which proved the reliability of VIGS in pepper. No significant difference in *CaSPDS* expression was observed between the TRV:00 and WT plants, but *CaSPDS* expression in the TRV:*CaSPDS* plants was 79.6% lower than *CaSPDS* expression in the WT plants. (Appendix A).

Under normal temperature conditions, the phenotypes of TRV:00 and TRV:*CaSPDS* showed no significant difference. After 4 °C treatment for 12 h, the TRV:*CaSPDS* seedlings showed mild leaf wilt. After 4 °C treatment for 24 h and 0 °C treatment for 15 min, the leaves of pepper showed obvious cold injury, and the TRV:*CaSPDS* plants showed more serious injury than the TRV:00 plants (Figure 5A).

3,3′-Diaminobenzidine (DAB) and nitro blue tetrazolium (NBT) staining can be used to determine the accumulation of H_2_O_2_ and O_2_^−^, respectively, which are the main reactive oxygen species (ROS). The leaves of the TRV:*CaSPDS* plants displayed more staining spots than those of the TRV:00 plants (Figure 5B). Consistently, the relative electrolyte conductivity (REC) and malondialdehyde (MDA) content of the TRV:*CaSPDS* plants were significantly higher than those of the TRV:00 plants after combined treatment at 0 °C and 4 °C (Figure 5C,D). The TRV:*CaSPDS* plants had significantly lower contents of proline (Pro), an osmoregulation substance, than the TRV:00 plants (Figure 5E). The differences in the above physiological indices under cold stress prompted us to study the activities of superoxide dismutase (SOD), peroxidase (POD), and hydrogen peroxidase (CAT) in pepper seedlings. After cold treatment (4 °C) for 12 h, the SOD and CAT activities of the *CaSPDS*-silenced plants were significantly higher than those of the control plants. However, after combined cold treatment (0 °C and 4 °C), the activities of these three antioxidant enzymes were significantly lower in the *CaSPDS*-silenced plants than in the control plants (Figure 5F–H). 

### 2.6. CaSPDS-Overexpressing (CaSPDS-OE) Arabidopsis

Based on the qRT-qPCR results of the homozygous T3 transgenic OE lines, we selected three lines (OE-3, OE-4, and OE-8) with the highest expression levels of *CaSPDS* for subsequent experiments (Appendix A). Under normal conditions and chilling stress (4 °C), no obvious difference was observed between the *CaSPDS*-OE lines and WT plants. After freezing stress (−8 °C), the leaves of most WT plants were obviously frostbitten, with phenotypes such as wilting and colour deepening, whereas the leaves of some OE lines showed frostbite. Three days after the freezing stress was removed, the WT plants had a low survival rate of 38%, whereas the OE lines had a survival rate of 62–70% (Figure 6A,B). We also measured the RECs, MDA contents, and SOD activities of the WT plants and OE lines (Figure 6C,D). Consistent with the above phenotype, no significant difference was found between the OE lines and WT plants under normal temperature and chilling stress. Under freezing stress, the RECs and MDA contents in the OE lines were significantly lower than those in the WT plants, whereas the SOD activities in the OE lines were significantly higher than those in the WT plants. These results demonstrate that *CaSPDS* confers freezing tolerance in transgenic *Arabidopsis*.

### 2.7. Determination of PA Content in Arabidopsis 

To understand the roles of PAs in the *CaSPDS*-OE lines under cold stress, we measured the contents of Put, Spd, and Spm (Figure 7A–C). No significant difference in Put content was found between the WT and OE lines under the control condition, but the Spd and Spm contents were significantly higher in the OE lines than in the WT plants. After chilling stress, the Put and Spd contents in the WT plants and OE lines increased, whereas the Spm content showed the opposite trend. After freezing stress, significant differences in Put, Spd, and Spm contents were found between the WT and OE lines. The Put and Spm contents in the OE lines were significantly lower than those in the WT plants, but the Spd contents in the OE lines were significantly higher than those in the WT plants. The above results showed that SPDS can catalyse Put to synthesise a large amount of Spd, thereby improving the freezing resistance of transgenic lines.

### 2.8. Expression of COR Genes in Arabidopsis

The expression levels of four COR genes were measured using qRT-PCR to explore the molecular mechanisms by which *CaSPDS* confers cold resistance in transgenic lines. Under control conditions, the expression levels of COR genes (*AtCOR15A*, *AtRD29A*, *AtCOR47*, and *AtKIN1*) in the WT and OE lines were relatively low with no significant difference between them. However, after freezing stress, the expression levels of these genes were significantly higher in the OE lines than in the WT plants (Figure 8A–D). These results further indicate that *CaSPDS* overexpression can positively regulate COR genes (*AtCOR15A*, *AtRD29A*, *AtCOR47*, and *AtKIN*), thus enhancing the freezing resistance of transgenic OE lines.

## 3. Discussion

Spd is a common PA that plays an important role in regulating plant growth and development [30,31]. It is also involved in responses to environmental stress [32,33]. The synthesis of Spd is affected by substrate availability and SPDS activity. At present, the gene encoding SPDS has been cloned in many species, such as cherry [26], apple [22], tobacco [20], eggplant [21], gentian [18], soybean [27], *Arabidopsis* [34], scots pine [35], tomato [19], and rice [23]. However, the roles of SPDS in pepper remain unclear. In the present study, we identified and isolated an *SPDS* gene (*CaSPDS*) from pepper and found that it contains two conservative domains (Figure 2B), which play important roles in Spd synthesis. Consistent with the results of previous studies [21], the phylogenetic analysis in the present study showed that *CaSPDS* has higher homology with the *SPDS* from Solanaceae plants than with the *SPDS* from Brassicaceae plants, such as *Arabidopsis*, oilseed rape, and *Camelina*.

Previous studies found that the expression of *SPDS* in plants is tissue specific. For example, *CpSPDS* expression is high in alabastrums and fruitlets but low in mature leaves [26]. Gomez-Jimenez et al. [36] reported that *OeSPDS* expression is high in the leaves, flowers, and fruits of olive trees but low in the shoots. In the present study, *CaSPDS* was highly expressed in the flowers and mature fruits of pepper plants but not in the cotyledons (Figure 3A). This result implies that *CaSPDS* is involved in regulating the formation of flower organs and fruit ripening. In addition, *CaSPDS* expression sharply increased under cold stress (Figure 3B,C), which is consistent with the findings of Kasukabe et al. [24] on sweet potato. This result also indicates that *CaSPDS* plays an important role in the response of pepper seedlings to cold stress.

Subcellular localisation revealed that CaSPDS protein is located in the nucleus (Figure 4B). This result is consistent with previous findings on AtSPDS1 and StSPDS [26,37]. By contrast, AtSPDS2 and CpSPDS are specifically localised in the nucleus and cytoplasm [21,37]. The difference in SPDS localisation may be owing to the variation between species. We therefore constructed an expression vector containing *CaSPDS* and conducted prokaryotic expression analysis. Kasukabe et al. [24,25] found through western blot analysis that the overexpression gene (*FSPD1*) has a translation protein in transgenic plants. Previous researchers obtained SPDS recombinant proteins of 34.9 and 38.7 kDa from gentian and *Nicotiana sylvestris*, respectively, and confirmed that they still exert catalytic activities [18,38]. Vuosku et al. [35] obtained a molecular mass of 55.5 kDA (including fusion tag) for the protein encoded by PsSPDS, and found that purified PsSPDS catalyses the synthesis of both Spd and Spm. In the present study, we obtained a soluble 37.59 kDa protein under the optimal induction conditions. This result provides a basis for studying the functions of CaSPDS protein in the future.

VIGS act as an important reverse genetic tool, which was used to silence genes after transcription, resulting in lower expression level of target genes or loss of their functions [39]. To explore the roles of *CaSPDS* in cold stress response, we silenced this gene in pepper plants through VIGS. Results showed that the *CaSPDS*-silenced plants were more sensitive to cold stress than the WT plants (Figure 5A). ROS, such as H_2_O_2_ and O_2_^−^, are inevitable by-products of aerobic metabolism. Under cold stress, a small amount of ROS interacts with Ca^2+^, phosphatidic acid, and other target proteins as signalling molecules to participate in cold stress response [40,41,42]. However, large amounts of ROS act as toxic substances, leading to plant cell dysfunction and even death [43]. In the present study, the larger accumulation of H_2_O_2_ and O_2_^−^ in TRV:*CaSPDS* plants than in TRV:00 plants reduced cold tolerance, which was confirmed by the increased contents of Pro and MDA (Figure 5C,D). To scavenge excess ROS during environmental stress, plants have evolved antioxidant defence systems, including antioxidant enzymes SOD, POD, and CAT [44,45,46]. After treatment with 4 °C for 12 h, the SOD and CAT activities in the TRV:*CaSPDS* plants were higher than those in the TRV:00 plants. However, after combined treatment (0 °C and 4 °C), the activities of antioxidant enzymes in these two materials showed opposite trends. These results suggest that the TRV:*CaSPDS* plants need higher antioxidant enzyme activities to alleviate the damage caused by mild cold stress. When the processing time was extended, causing severe stress, the large amounts of ROS that accumulated after *CaSPDS* silencing could not be scavenged, causing serious damage to plants.

*CaSPDS* was overexpressed in *Arabidopsis* and then subjected to chilling (4 °C) and freezing (−8 °C) to further confirm the roles of *CaSPDS* in plant tolerance against cold stress. Compared with that of the WT plants, the freezing resistance of the *CaSPDS*-OE lines was significantly stronger (Figure 6A). Correspondingly, the survival rate of the transgenic lines was 66.67%, while that of the WT plants was only 38% (Figure 6B).

To adapt to adverse environments, plants adjust at the physiological and molecular levels. Neily et al. [22] found that the REC and antioxidant enzyme activity were related to the salt tolerance of plants heterologously expressing *MdSPDS1*. Similarly, the antioxidant enzyme activities of *FSPD1*-transgenic sweet potato were found to be higher than those of WT plants, indicating the greater chilling resistance of transgenics than WT plants [24]. Yang et al. [47] found that the overexpression of the *SPDS* gene in potato promoted PAs content and antioxidant enzyme activity, thus improving the cold resistance of transgenic plants. In the present study, the *CaSPDS*-OE lines reduced the MDA content induced by freezing stress (Figure 6D). In addition, the PA contents in the WT and transgenic plants differed. Among them, the Put and Spm contents of the *CaSPDS*-OE lines significantly decreased, indicating that *CaSPDS* overexpression promoted the synthesis of Put to Spd but not the synthesis of Spd to Spm (Figure 7). This phenomenon led to the accumulation of a large amount of Spd, which improved the freezing tolerance of the transgenic plants. Similar results were obtained by Kasukabe et al. [25]. The expression levels of COR genes *AtCOR15A*, *AtRD29A*, *AtCOR47*, and *AtKIN1* in the transgenic lines were also significantly higher than those in the WT plants (Figure 8). Our analysis of the functions of *CaSPDS*, including measurement of gene expression (qRT-PCR and RNA-seq), gene silencing in pepper, and gene overexpression in *Arabidopsis*, thus suggests that *CaSPDS* positively regulates the cold stress response of pepper. However, further investigation is required to determine the specific molecular mechanism of cold tolerance mediated by *CaSPDS*.

## 4. Materials and Methods

### 4.1. Plant Material and Treatments

Pepper (*Capsicum annuum* L.) cultivar ‘Gan zi’ was used in the experiments. Seeds were soaked in warm water (55 °C, 15 min and 25 °C, 6 h) to promote germination and placed on moist filter paper in a Petri dish at 28 °C. Germinated seeds were selected and sown in a plug containing a mixture of perlite:vermiculite:peat (*v*/*v*/*v*/ = 1:1:2) and placed in a growth chamber under 25 °C light (16 h)/20 °C dark (8 h) conditions until 6–8 true leaves appeared. The seedlings were then subjected to chilling stress (4 °C) for 0, 3, 6, 12, and 24 h. For tissue-specific expression analysis, ten tissues, including young leaf (YL), young stem (YS), young root (YR), cotyledon (C), flower (F), young fruit (YF), mature fruit (MF), mature leaf (ML), mature stem (MS), and mature root (MR) were collected, as previously described [48]. For the treatment of silenced plants, TRV:00 and TRV:*CaSPDS* plants were treated at 4 °C for 12 h, placed at 0 °C for 15 min, and then treated at 4 °C again for 12 h. Samples were collected 0, 12, and 24 h after treatment. All samples were immediately placed in liquid nitrogen and then stored at −80 °C.

*Arabidopsis* ecotype Columbia (Col-0) was used for gene transformation in this experiment. Seeds of transgenic lines and WT were soaked in water, vernalised at 4 °C for 3 days, and then sown in a high-pressure-sterilised nutrient mixture. The matrix formulation and culture conditions were consistent with those mentioned earlier. After growing for 3–4 weeks, the *Arabidopsis* seedlings were subjected to chilling (4 °C, 24 h) and freezing (−8 °C, 6 h) stress. After 3 days of recovery at normal temperature, the survival rates of stressed plants (after freezing stress) were calculated.

### 4.2. Bioinformatics Analysis of CaSPDS

The physicochemical properties of CaSPDS protein, including protein length (amino acids), protein weight (kDa), isoelectric point (PI), and instability index, were obtained using ExPASy (http://web.expasy.org/protparam/ (accessed on 17 May 2022)). The gene structure of *CaSPDS* was analysed using GSDS (http://gsds.cbi.pku.edu.cn/, accessed on 17 May 2022). The secondary and tertiary structures of CaSPDS protein were analysed using the NPSA-PRABI website (https://npsa-prabi.ibcp.fr/cgi-bin/npsa_automat.pl?page=npsa%20_sopma.html (accessed on 21 May 2022)) and SWISS-MODEL website (https://swissmodel.expasy.org/interactive (accessed on 21 May 2022)). The hydrophilic and hydrophobic regions of the protein were analysed using the ProtScale website (http://web.expasy.org/protscale (accessed on 2 June 2022)), and transmembrane domains were obtained using TMHMM 2.0 (http://www.cbs.dtu.dk/services/TMHMM/ (accessed on 2 June 2022)). The signal peptides were predicted using the SignalP 6.0 program (https://services.healthtech.dtu.dk/service.php?SignalP (accessed on 2 June 2022)). The predicted phosphorylation sites (serine, threonine, and tyrosine) were analysed using NetPhos 3.1 (http://www.cbs.dtu.dk/services/NetPhos (accessed on 2 June 2022)).

### 4.3. Phylogenetic Analysis and Multiple Sequence Alignment of SPDS Proteins

The full-length proteins of 16 species, including pepper (NP_001311579.1), potato (XP_006340116.1), tomato (NP_001234493.1 and XP_004237263.1), petunia (BAC55523.1); common tobacco (XP_016439736.1), woodland tobacco (NP_001289514.1), *Arabidopsis* (AT1G23820.1 and AT1G70310.1), gentian (BAS21156.1), lettuce (XP_023755202.1), soybean (XP_006579563.1), jujuba (XP_015894683), clementine (CAO02391.1), oilseed rape (XP_013697453.2), maize (NP_001149310.1), and rice (NP_001389979.1), were aligned using ClustalW with default parameters. MEGA-X software was used to construct a phylogenetic tree using the maximum likelihood method and pairwise deletion with 1000 bootstrap replicates. SPDS protein sequences of *Arabidopsis*, pepper, potato, tomato, petunia, common tobacco, and woodland tobacco were aligned using ClustalX software (1.83) with default parameters to clarify the evolutionary relationship between CaSPDS in pepper, AtSPDS in *Arabidopsis*, and SPDSs of other Solanaceae species [49].

### 4.4. Gene Expression Analysis

Total RNA was isolated from *Arabidopsis* and pepper samples using the RNAprep Pure Plant Plus Kit (TIANGEN, Beijing, China), and the concentration was detected using a spectrophotometer (Thermo Fisher Scientific Oy, Finland). First-strand cDNA was synthesised using the PrimeScript™ RT reagent Kit (TaKaRa, Dalian, China) in accordance with the manufacturer’s instructions. qRT-PCR was performed using the CFX96 real-time PCR system (Bio-Rad, Hercules, CA, USA) with 2 × SYBR Green Fast qPCR Mix (Biomarker, China). The relative expression of *CaSPDS* was calculated using the 2^−∆∆Ct^ method [50]. *Atactin2* (AT5G09810) and *CaUbi3* (LOC107873556) were used as the reference genes for *Arabidopsis* and pepper, respectively [42,51]. The primers used for qRT-PCR are listed in Appendix A.

### 4.5. Construction of Cloning Vector

The sequence of *CaSPDS* was amplified via PCR using PrimeSTAR Max DNA Polymerase (TaKaRa, Dalian, Chian). After purification, the amplified product was connected into the pEASY-Blunt Simple Cloning Vector (TransGen Biotech, Beijing, China) and then sent to the company for sequencing to confirm. All primers are listed in Appendix A.

### 4.6. Subcellular Localisation

The open reading frame (ORF) sequence of *CaSPDS* was cloned into the pBWA(V)HS-GLosgfp vector using the BsaⅠ and Eco3IⅠ restriction endonucleases to form pBWA(V)HS-*CaSPDS*-GLosgfp. This fusion construct was transformed into tobacco epidermal cells using *Agrobacterium*-mediated transient transformation [52]. pBWA(V)HS-GLosgfp was used as a control. After 36–72 h of darkness, a Nikon C2-ER confocal laser scanning microscope (Nikon Instruments, Tokyo, Japan) was used to collect fluorescence images.

### 4.7. Expression and Purification of CaSPDS Protein

The sequence of *CaSPDS* was obtained from the cloning vector by using EcoRI and SaI1 restriction endonucleases (TransGen Biotech, Beijing, China) and then connected into pCold I vector (TaKaRa, Tokyo, Japan) by T4 ligase (TransGen Biotech, Beijing, China), producing pCold-*CaSPDS*. The pCold I empty vector and validated pCold-*CaSPDS* were transfected into the *Escherichia coli* BL21 (DE3) cells (TaKaRa, Dalian, China) and then cultured in Luria–Bertani (50 µg/mL Amp) liquid medium at 37 °C with shaking at 200 rpm until the OD600 reached 0.6. Different concentrations of isopropyl-β-d-thiogalactoside (IPTG, 0, 0.2, 0.4, 0.6, 0.8, and 1.0 mM) were added for 24 h to induce expression. It was then induced at the optimum IPTG concentration for 0, 2, 6, 12, 24, and 36 h. The ultrasonic fragmentation and purification of the CaSPDS recombinant protein were conducted as previously reported [48].

### 4.8. VIGS of Pepper Seedlings

For VIGS, the coding regions of *CaSPDS* and *CaPDS* (phytoene dehydrogenase) were connected into a pTRV2 vector (Fenghui Biotechnology, Changsha, China) using XbaI and KpnI restriction endonucleases to generate the constructs pTRV2:*CaSPDS* and pTRV2:*CaPDS*. Then, pTRV1 (Fenghui Biotechnology, Changsha, China), pTRV2:00 (negative control), pTRV2:*CaSPDS* (positive control), and pTRV2:*CaPDS* were transformed into *Agrobacterium tumefaciens* strain GV3101. *CaSPDS* expression was silenced in pepper following the method described by Velásquez et al. [53]. At 25 days post-infiltration, the silencing efficiency of the pTRV2:*CaSPDS* plants was calculated, and plants with rates higher than 50% were used for follow-up tests.

### 4.9. Arabidopsis Transformation

The ORF regions of *CaSPDS* were cloned into the pBWA(V)HS vector to form the pBWA(V)HS-*CaSPDS* recombinant vector and then transformed into *Agrobacterium tumefaciens* strain GV3101, and the floral-dip method was used for Arabidopsis transformation. The homozygous transgenic *Arabidopsis* plants were obtained as previously described [48].

### 4.10. DAB and NBT Staining

DAB and NBT staining were used to detect the accumulation of H_2_O_2_ and O_2_^−^ in plants, respectively [54,55]. Fresh leaves of pepper were soaked in dyeing liquid for 8–10 h in the dark, immersed in 95% ethanol, and then heated in a boiling water bath until they faded to white. The staining solutions for H_2_O_2_ and O_2_^−^ were 5 mM DAB and 0.75 mM NBT, respectively. Each group of three leaves was repeated three times.

### 4.11. Determination of Polyamine Content

We homogenized 0.5 g frozen leaves homogenized in 3 mL of 5% perchloric acid and incubated in ice-water for 1 h, followed by centrifugation at 13,000× *g* for 25 min at 4 °C. Volumes of 500 µL of the supernatant were mixed with 1 mL of 2 M NaOH and 10 µL of benzoyl chloride in a plastic tube, and incubated in darkness for 20 min at 37 °C. The reaction was quenched by addition 2 mL of saturated NaCl and 2 mL of dimethyl ether and further centrifuged at 5000× *g* for 8 min. One milliliter of the ether phase was evaporated to dryness. PAs were dissolved in 1 mL of methanol and filtered with a 0.45 µm pore nylon filter. The injection volume was 10 µL. The mobile phase consisted of methanol (70%) and water (30%), with a flux of 0.7 mL/min. The PA peaks were detected at 230 nm. The standard calibration curves were constructed according to Lin et al. [56]. The standard sample was purchased from Sigma-Aldrich (Dallas, TX, USA).

### 4.12. Determination of Physiological Indices in Plants

The REC was measured using the immersion method [57]. Pro content was assayed following the methods described by Bates et al. [58]. Briefly, 0.5 g leaf samples were added in 5 mL of 3% sulfosalicylic acid and then placed in boiling water for 10 min. After filtering, the supernatant (2 mL) was mixed with 2 mL of acetic acid glacial and 2 mL of ninhydrin. Subsequently, the mixture was maintained in boiling water for 30 min. After cooling, 4 mL of methylbenzene was added. Absorbance was measured at 520 nm using methylbenzene as blank. MDA content was determined using a plant MDA assay kit (Nanjing Jiancheng, Nanjing, China) in accordance with the manufacturer’s instructions. The activities of antioxidant enzymes (SOD, POD, and CAT) were determined following previously described procedures [48].

### 4.13. Statistical Analysis

Statistical analysis was performed using SPSS 17.0 (IBM Corp., Armonk, NY, USA), and the means were compared using Duncan’s test. Significant differences and highly significant differences were considered at *p* < 0.05 and *p* < 0.01, respectively.

## 5. Conclusions

We isolated and identified an *SPDS* gene from *C. annuum*, *CaSPDS*, which contained two highly conserved domains and was localised in the nuclei. Expression analysis of *CaSPDS* indicated that it was highly expressed in the stems, flowers, and mature fruits of pepper and was rapidly induced by cold stress. Functional verification of *CaSPDS* was performed by silencing this gene in pepper and overexpressing it in *Arabidopsis*. The *CaSPDS*-silenced plants showed more serious cell damage, more ROS accumulation, and reduced tolerance to cold stress than the WT plants. Under cold stress, the *CaSPDS*-OE *Arabidopsis* seedlings had higher survival rates than the WT seedlings. The above results indicate that *CaSPDS* is valuable in regulating the cold tolerance of pepper.

## Figures and Tables

**Figure 1 ijms-24-05013-f001:**
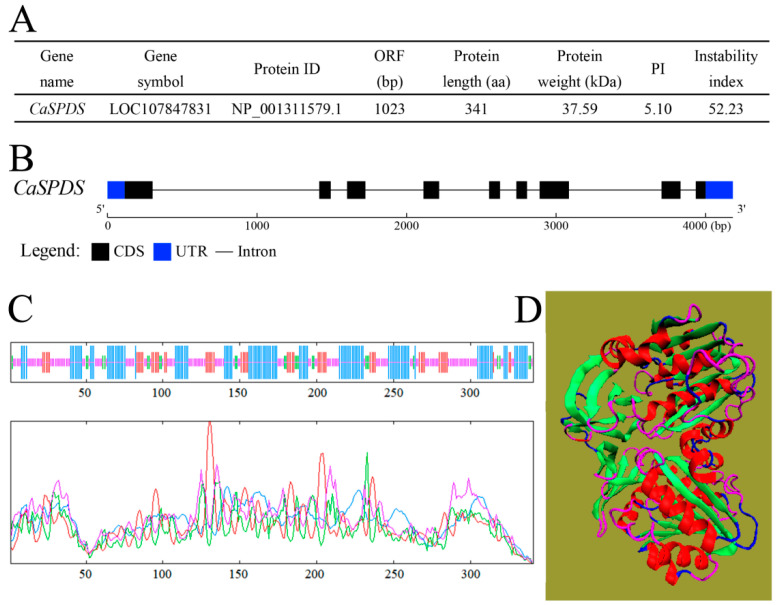
Basic information about *CaSPDS* sequence. (**A**) Detailed physicochemical properties of *CaSPDS*. (**B**) Genomic structure of *CaSPDS*. Blue rectangles denote upstream/downstream sequences; black rectangles and lines denote exons and introns, respectively. (**C**) Predicted secondary structure of CaSPDS. Numbers represent the number of amino acids. (**D**) Predicted tertiary structure of CaSPDS. Red denotes the alpha helices, green extended strands, purple beta turns, and blue random coils.

**Figure 2 ijms-24-05013-f002:**
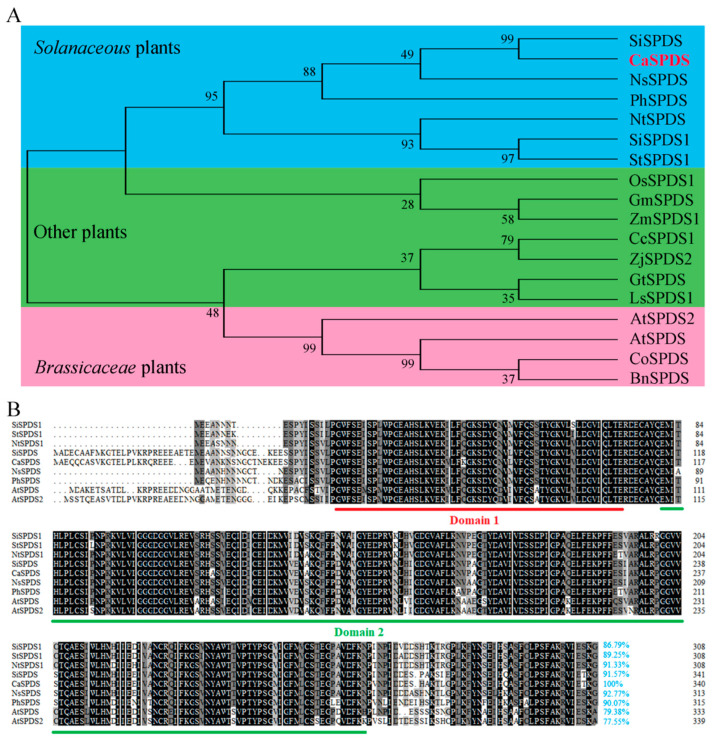
Phylogenetic tree and multiple sequence alignment of SPDS proteins. (**A**) Phylogenetic analysis of plant SPDS proteins. Red characters represent the SPDS protein in pepper. Blue background denotes *Solanaceae* plants, pink background denotes *Brassicaceae* plants, and green background denotes other plants. St: potato (*Solanum tuberosum* L.); Si: tomato (*Solanum lycopersicum* L.); Ph: petunia (*Petunia hybrida* Vilm.); Nt: common tobacco (*Nicotiana tabacum* L.) Ns: woodland tobacco (*Nicotiana sylvestris* L.); At: Arabidopsis (*Arabidopsis thaliana*); Gt: gentian (*Gentiana triflora* ‘SpB’); Ls: lettuce (*Lactuca sativa* Linn.); Gm: soybean (*Glycine max* L.); Zj: jujuba (*Ziziphus jujuba* Mill.); Cc: clementine (*Citrus clementina*); Zm: maize (*Zea mays* L.); Os: rice (*Oryza sativa* L.); Bn: oilseed rape (*Brassica napus* L.); Co: camelina (*Cochlearia officinalis* L.). (**B**) Alignment of *Arabidopsis* and Solanaceae SPDS sequences. Domain1 (spermidine synthase tetramerisation domain) was labelled with red line, and domain2 (spermine/spermidine synthase domain) was labelled with green lines.

**Figure 3 ijms-24-05013-f003:**
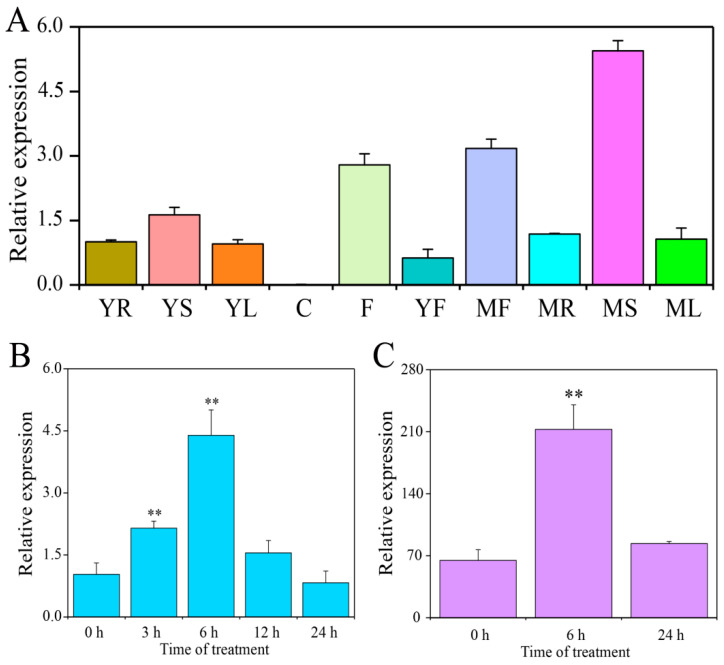
Expression analysis of *CaSPDS* in various tissues and in response to cold stress. (**A**) qRT-PCR analysis of *CaSPDS* in ten tissues: young root (YR); young stem (YS); young leaf (YL); cotyledon (C); flower (F); young fruit (YF); mature fruit (MF); mature root (MR); mature stem (MS); mature leaf (ML). Data are expressed relative to YR and normalised to the reference gene (CaUbi3, AY486137). (**B**) Expression profiling of *CaSPDS* in response to cold stress by qRT-PCR. Pepper seedlings were subjected to cold stress at 4 °C. Leaves were sampled at 0, 3, 6, 12, and 24 h after treatment. (**C**) Expression patterns of *CaSPDS* in response to cold stress using available RNA-seq data. Pepper leaves were collected at 0, 6 and 24 h after 4 °C treatment. ‘**’ indicates a significant difference between the treatment and control at *p* < 0.01.

**Figure 4 ijms-24-05013-f004:**
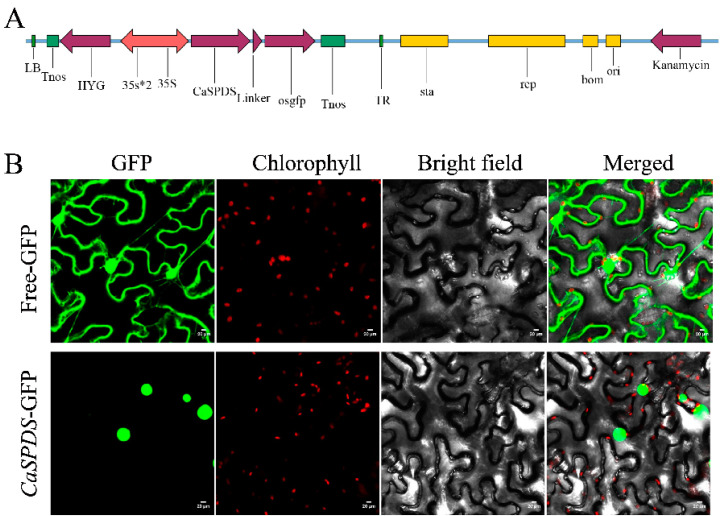
Subcellular localisation of CaSPDS. (**A**) pBWA(V)HS-*CaSPDS*-GLosgfp fusion vector construction. (**B**) Subcellular localisation of CaSPDS in tobacco leaves. Localisation of GFP and its fusion proteins is shown in green, and the chloroplast fluorescence is shown in red. Scale bars = 20 µm.

**Figure 5 ijms-24-05013-f005:**
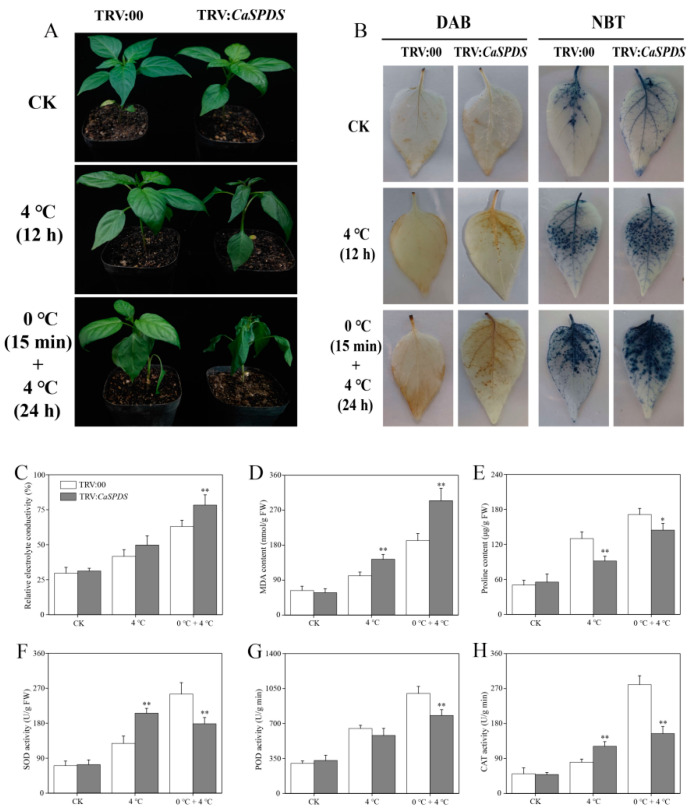
Effect of *CaSPDS* silencing on pepper tolerance to cold stress. (**A**) Phenotype, (**B**) tissue staining, (**C**) relative electrolytic leakage, (**D**) MDA content, (**E**) proline content, (**F**) SOD activity, (**G**) POD activity, and (**H**) CAT activity of TRV:00 and TRV:*CaSPDS* plants after cold treatment. “*” represents *p* < 0.05, and “**” represents *p* < 0.01 between WT and transgenic lines.

**Figure 6 ijms-24-05013-f006:**
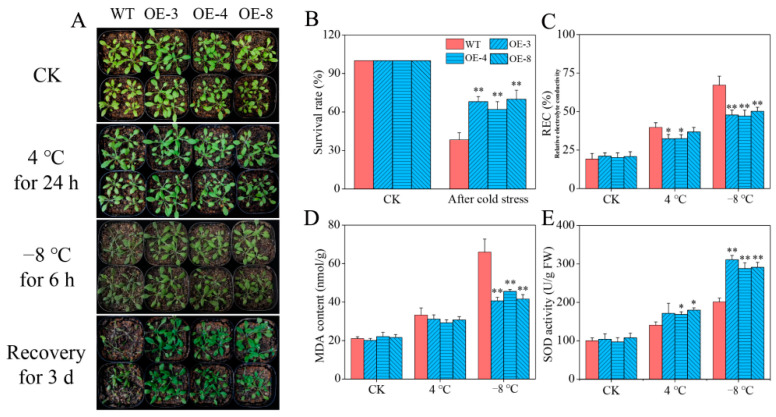
Effect of *CaSPDS* overexpression on *Arabidopsis* tolerance to cold stress. (**A**) Phenotype, (**B**) survival rate, (**C**) relative electrolytic leakage, (**D**) MDA content, and (**E**) SOD activity of WT and transgenic lines after cold stress. “*” represents *p* < 0.05, and “**” represents *p* < 0.01 between WT and transgenic lines.

**Figure 7 ijms-24-05013-f007:**
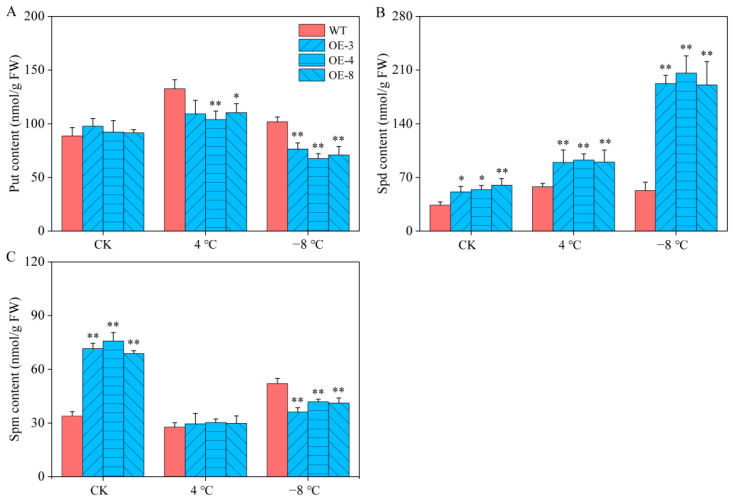
PA content in pepper under cold stress. (**A**) Put content. (**B**) Spd content. (**C**) Spm content. “*” represents *p* < 0.05, and “**” represents *p* < 0.01 between WT and transgenic lines.

**Figure 8 ijms-24-05013-f008:**
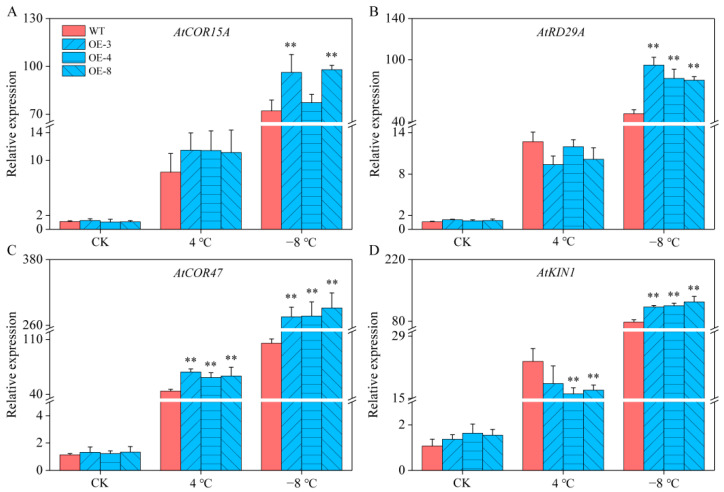
Expression levels of (**A**) *AtCOR15A*, (**B**) *AtRD29A*, (**C**) *AtCOR47*, and (**D**) *AtKIN1* of WT and *CaSPDS*-OE plants under cold stress. “*” represents *p* < 0.05, and “**” represents *p* < 0.01 between WT and transgenic lines.

## Data Availability

Not applicable.

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
