# Peer review of "CaSPDS, a Spermidine Synthase Gene from Pepper (Capsicum annuum L.), Plays an Important Role in Response to Cold Stress"

_ijms, 2023, doi:10.3390/ijms24055013_

Round 1

Reviewer 1 Report

This manuscript presents a study on the function of CaSPDS, a SPDS gene, isolated from pepper, in response to cold stress by silencing and overexpressing this gene in pepper and Arabidopsis. In general, the study is well-designed, the results are stated concisely and supported by the bioinformatics analysis, qRT-PCR and functional assays. The findings contribute to understanding of the role of SPDS genes in the regulation of plant responses to environmental stresses and would be of interest to the IJMS readers. However, prior to publication, some corrections and additional clarifications are necessary to improve the overall quality and scientific rigour of the manuscript. These changes include adding more details to some methods, providing further explanation for some results to enhance clarity.

Page 3, lines 89-90: Instead of "..identified in the pepper genome by BLAST sequencing", use "identified in the pepper genome through a BLAST sequence analysis". BLAST is used to compare a sequence of interest with the sequences present in a database and identify similarities, but not for sequencing.  

Page 3, lines 94-95: Replace “… was located at 11 chromosomes”, with “…located on the 11th chromosome”

Page 5, lines 137-141: The statement “We examined the expression of CaSPDS in 10 tissue samples of pepper in different stages, including seedling, flowering (flower), and fruiting” lacks clarity. Please rephrase the text to better convey that various stages and seedling parts were sampled.

Page 5, lines 146, 148, and 158: The authors repeatedly refer to "transcriptome data", citing the work of Franceschetti et al. (2004). Transcriptome data typically refers to the complete set of RNA transcripts that are produced by the genome, which are usually obtained using high-throughput sequencing and computational analysis. However, it seems that the cited paper does not actually provide such transcriptome data. What is the intended meaning of the "transcriptome data" phrase?

Page 6, line 161: For greater clarity on the injection site, I suggest using "was constructed and injected into tobacco leaves" rather than simply "tobacco".

The authors must improve the Materials and Methods section, providing more specific and detailed information about the experimental design, procedures, and data analysis methods. 

Page 13, lines 419-421: The relative expression of CaSPDS was calculated using the 2−ΔΔCt method with Actin2 (AT5G09810) and CaUbi3 (LOC107873556) as the reference genes for Arabidopsis and pepper, respectively. Did the authors use only a single reference gene for the calculation of relative expression? It is generally recommended to use at least two reference genes to ensure accurate normalisation of the data.

Page 14, lines 463-464: In the section, describing Arabidopsis transformation, it is stated …then transformed into GV3101”. It must be at least mentioned that they used Agrobacterium tumefaciens to transform plants and specify the method, e.g. floral  dip transformation.  

Page 14, line 468: While the authors reference Velásquez et al. and Alvarez et al. when describing DAB and NBT staining, there is no mention of NBT staining in these papers. They should provide additional details in M&M section or suggest a more appropriate source for the NBT staining protocol.

Page 14, line 483: The authors should provide additional details about the methodology used to detect the content of polyamines. Merely citing a reference is not sufficient to convey the necessary information about the technique employed.

Minors:

- The abbreviations must be introduced during the first mentioning in the text, e.g. SOD, POD, CAT, etc.

- Gene names should be italicised: lines 157, 251, 253, 346, etc.

The manuscript presents well-supported conclusions based on the data provided in the text, figures, images and tables. Addressing the concerns outlined in the comments would further strengthen the study and bring it up to the standards required for acceptance and publication in IJMS.

Author Response

Please see the attachment:

Response to Reviewer 1 Comments

This manuscript presents a study on the function of CaSPDS, a SPDS gene, isolated from pepper, in response to cold stress by silencing and overexpressing this gene in pepper and Arabidopsis. In general, the study is well-designed, the results are stated concisely and supported by the bioinformatics analysis, qRT-PCR and functional assays. The findings contribute to understanding of the role of SPDS genes in the regulation of plant responses to environmental stresses and would be of interest to the IJMS readers. However, prior to publication, some corrections and additional clarifications are necessary to improve the overall quality and scientific rigour of the manuscript. These changes include adding more details to some methods, providing further explanation for some results to enhance clarity.

Page 3, lines 89-90: Instead of "..identified in the pepper genome by BLAST sequencing", use "identified in the pepper genome through a BLAST sequence analysis". BLAST is used to compare a sequence of interest with the sequences present in a database and identify similarities, but not for sequencing.  

Thank you very much for your comment. We have revised the ‘..identified in the pepper genome by BLAST sequencing’ to ‘identified in the pepper genome through a BLAST sequence analysis’. The details were in the manuscript by marking in red.

Page 3, lines 94-95: Replace “… was located at 11 chromosomes”, with “…located on the 11th chromosome”

Thank you very much for your comment. We have revised the ‘… was located at 11 chromosomes’ to ‘…located on the 11th chromosome’. The details were in the manuscript by marking in red.

Page 5, lines 137-141: The statement “We examined the expression of CaSPDS in 10 tissue samples of pepper in different stages, including seedling, flowering (flower), and fruiting” lacks clarity. Please rephrase the text to better convey that various stages and seedling parts were sampled.

Thank you for your good suggestions of our manuscript. We have revised this sentence. The details were in the manuscript by marking in red.

Page 5, lines 146, 148, and 158: The authors repeatedly refer to "transcriptome data", citing the work of Franceschetti et al. (2004). Transcriptome data typically refers to the complete set of RNA transcripts that are produced by the genome, which are usually obtained using high-throughput sequencing and computational analysis. However, it seems that the cited paper does not actually provide such transcriptome data. What is the intended meaning of the "transcriptome data" phrase?

We are very sorry for our incorrect writing. Transcriptome data is the preliminary work of our laboratory and has been published in the NCBI database. We have revised this sentence. The details were in the manuscript by marking in red. 

Page 6, line 161: For greater clarity on the injection site, I suggest using "was constructed and injected into tobacco leaves" rather than simply "tobacco".

Thank you for your good suggestions of our manuscript. We have revised this sentence. The details were in the manuscript by marking in red.

The authors must improve the Materials and Methods section, providing more specific and detailed information about the experimental design, procedures, and data analysis methods. 

Page 13, lines 419-421: The relative expression of CaSPDS was calculated using the 2−ΔΔCt method with Actin2 (AT5G09810) and CaUbi3 (LOC107873556) as the reference genes for Arabidopsis and pepper, respectively. Did the authors use only a single reference gene for the calculation of relative expression? It is generally recommended to use at least two reference genes to ensure accurate normalisation of the data.

Thank you for your good suggestions of our manuscript. In this study, the reference genes were selected from six pairs of primers in pepper and four pairs of primers in Arabidopsis. The primer information is as follows:

Species

Gene name

Accession NO.

Forward primer (5’-3’)

Reverse primer (5’-3’)

Pepper

β-tubulin

EF495259

GAGGGTGAGTGAGCAGTTC

CTTCATCGTCATCTGCTGTC

CaActin-100

LOC107840006

CCACCTCTTCACTCTCTGCTCT

ACTAGGAAAAACAGCCCTTGGT

CaActin-100

LOC107840006

AGGGATGGGTCAAAAGGATGC

GAGACAACACCGCCTGAATAGC

CaActin-66

LOC107843797

GGTGACGAGGCTCAATCCAA

CTCTGGAGCCACACGAAGTT

Actin

GQ339766.1

AGCACCTCTCAACCCTAA

GCAAAGCATAACCCTCAT

CaUbi3

LOC107873556

TGTCCATCTGCTCTCTGTTG

CACCCCAAGCACAATAAGAC

Arabidopsis

Actin2

AT5G09810

TAACAGGGAGAAGATGACTCAGATCA

AAGATCAAGACGAAGGATAGCATGAG

AtActin

AT3G18780

TGGGTTTTTACTTACGTCTGCG

GGGAACAAAAGGAATAAAGAGGC

AtActin

AT3G18780

GGAAAGGATCTGTACGGTAAC

TGTGAACGATTCCTGGAC

AtActin

AT3G18780

GGTAACATTGTGCTCAGTGGTGG

AACGACCTTAATCTTCATGCTGC

Page 14, lines 463-464: In the section, describing Arabidopsis transformation, it is stated “…then transformed into GV3101”. It must be at least mentioned that they used Agrobacterium tumefaciens to transform plants and specify the method, e.g. floral  dip transformation.  

Thank you for your good suggestions of our manuscript. We have revised this sentence. The details were in the manuscript by marking in red.

Page 14, line 468: While the authors reference Velásquez et al. and Alvarez et al. when describing DAB and NBT staining, there is no mention of NBT staining in these papers. They should provide additional details in M&M section or suggest a more appropriate source for the NBT staining protocol.

We are very sorry for our incorrect writing. The VIGS was based on the method of Velásquez et al, while the staining of DAB and NBT refer to the methods of Alvarez et al and Wohlgemuth et al. We have revised this sentence. The details were in the manuscript by marking in red. 

Page 14, line 483: The authors should provide additional details about the methodology used to detect the content of polyamines. Merely citing a reference is not sufficient to convey the necessary information about the technique employed.

Thank you for your good suggestions of our manuscript. We have added the method of PAs determination. The details were in the manuscript by marking in red.

Minors:

- The abbreviations must be introduced during the first mentioning in the text, e.g. SOD, POD, CAT, etc.

Thank you for your good suggestions. We have added it. The details were in the manuscript by marking in red.

- Gene names should be italicised: lines 157, 251, 253, 346, etc.

Thank you very much for your comment. We have revised it. The details were in the manuscript by marking in red.

The manuscript presents well-supported conclusions based on the data provided in the text, figures, images and tables. Addressing the concerns outlined in the comments would further strengthen the study and bring it up to the standards required for acceptance and publication in IJMS.

Reviewer 2 Report

-This work deals with the behaviour of the spermidine synthase gene in pepper plants, in response to cold stress. Is a very interesting work, with important information and very complete for the used techniques. 

Two criticisms could be made in terms of methodology.

- As a complement to the silencing trials, the authors could have performed supplementation treatments with synthetic spermidine.

- The election of arabidopsis as a specie for the transgenic experiment to chilling stress is a wrong decision because arabidopsis is adapted to cold stress. However, this is not invalidated the conclusions of this work.

In addition, the authors should check the followings points: 

-In figure 2A, CaSPDS is more related to monocotyledons plant as rice and maize than other dicotyledons, but in figure 2A also CaSPDS is related to GmSPDS in the same valour as ZmSPDS. What is the author's explanation of this?.

- In Figure 3A this nomenclature is very confusing to read, please change it.

- In Figures 3B and C, the stage of the cold treatments should be clarified, to facilitate the reading.

- In Figures 6A, 6D and 7B, the legends -8°C is not complete and the symbol "-" is missing.

Author Response

Please see the attachment:

Response to Reviewer 2 Comments

-This work deals with the behaviour of the spermidine synthase gene in pepper plants, in response to cold stress. Is a very interesting work, with important information and very complete for the used techniques. 

Two criticisms could be made in terms of methodology.

- As a complement to the silencing trials, the authors could have performed supplementation treatments with synthetic spermidine.

Thank you for your good suggestions of our manuscript. First of all, we are very sorry that we did not use exogenous PAs to treat CaSPDS-silented plants. Before conducting this experiment, we studied the effect of exogenous PAs on pepper under cold stress, and found that spraying 0.5 mM Spd could effectively improve the cold resistance of pepper seedlings. It is very meaningful to perform treatments with synthetic Spd using CaSPDS-silented plants as materials, which can further explore the role of Spd and CaSPDS in response of pepper to cold stress. Based on your suggestion, we will focus on the cold response mechanism of CaSPDS-silented plants in the future, and explore the molecular mechanism of CaSPDS gene participating in cold response using molecular biological technology.

- The election of arabidopsis as a specie for the transgenic experiment to chilling stress is a wrong decision because arabidopsis is adapted to cold stress. However, this is not invalidated the conclusions of this work.

Thank you very much for your comment. Through the pre-experiment, we observed that Arabidopsis is resistant to chilling stress, therefore, we added freezing stress under - 8 ℃ and the phenotype and survival rate of WT and transgenic lines were different. For cold stress, we will try to use pepper or model plants such as tomato and tobacco as materials for genetic transformation.

In addition, the authors should check the followings points: 

-In figure 2A, CaSPDS is more related to monocotyledons plant as rice and maize than other dicotyledons, but in figure 2A also CaSPDS is related to GmSPDS in the same valour as ZmSPDS. What is the author's explanation of this?.

Thank you very much for your comment. First, in order to verify the reliability of the evolutionary tree, we constructed the phylogenetic tree using NJ method with 1000 bootstrap replicates, and the results are shown in Figure 1. The CaSPDS was clustered in the same group with SPDS from other Solanaceae family. However, the pepper CaSPDS protein was more closely related to those from rice and maize (monocotyledons) than to those of Brassicaceae plants (dicotyledons), which was consistent with the results of Qiu et al[1]. In addition, in general, the gene family members of monocotyledonous plants will be clustered together[2-4]. However, for the SPDS gene family, the maize ZmSPDS are more colsely related to those of soybean than those to rice, which may be due to the fact that the numbers of SPDS is small and the gene family has not been expanded, we speculate that these SPDS genes were established prior to the divergence of monocotyledonous and dicotyledonous.

Figure 1. Phylogenetic analysis of CaSPDS and other SPDSs. The phylogenetic tree was constructed by using the NJ method with 1000 bootstrap replicates.

[1] Qiu, Z.; Yan, S.; Xia, B.; Jiang, J.; Yu, B.; Lei, J.; Chen, C.; Chen, L.; Yang , Y.; Wang, Y.; Tian, S.; Cao, B. The eggplant transcription factor MYB44 enhances resistance to bacterial wilt by activating the expression of spermidine synthase. J. Exp. Bot. 2019, 70, 5343-5354

[2] Yu, Y.; Yu, M.; Zhang, S.; Song, T.; Zhang, M.; Zhou, H.; Wang, Y.; Xiang, J.; Zhang, X. Transcriptomic Identifification of Wheat AP2/ERF Transcription Factors and Functional Characterization of TaERF-6-3A in Response to Drought and Salinity Stresses. Int. J. Mol. Sci. 2022, 23, 3272.

[3] Zhang, J.; Liang, L.; Xiao, J.; Xie, Y.; Zhu, L.; Xue, X.; Xu, L.; Zhou, P.; Ran, J.; Huang, Z.; et al.

Genome-Wide Identifification of Polyamine Oxidase (PAO) Family Genes: Roles of CaPAO2 and CaPAO4 in the Cold Tolerance of Pepper (Capsicum annuum L.). Int. J. Mol. Sci. 2022, 23, 9999.

[4] Zhang YT, Li YL, He YW, Hu WJ, Zhang Y, Wang XR, Tang HR (2018) Identifcation of NADPH oxidase family members associated with cold stress in strawberry. FEBS Open Bio 8:593–605

- In Figure 3A this nomenclature is very confusing to read, please change it.

Thank you for your good suggestions of our manuscript. We have revised the Figure 3A. The details were in the manuscript by marking in red.

- In Figures 3B and C, the stage of the cold treatments should be clarified, to facilitate the reading.

Thank you for your good suggestions of our manuscript. We have added the description of treatment time. The details were in the manuscript by marking in red.

- In Figures 6A, 6D and 7B, the legends -8°C is not complete and the symbol "-" is missing.

Thank you very much for your comment. We have revised the Figure 6 and Figure 7.

Reviewer 3 Report

This manuscript deals with the CaSPDS in pepper plays an important and positive role in response to cold stress. This work provides new sight and opinion into the development of molecular breeding to enhance the cold tolerance of pepper. However, several points need to be improved. Below are comments for the authors to improve their manuscript.

Tables and figures:

Figure 1: Please replace “LOC107847831” with “LOC107847831”.

Figure 2: Figure 2B is unclear. Please supplement the analysis results of phylogeny.

Figure 3: Please replace CaSPDS with CaSPDS, which is also need to be check in the text, such as Figure 4, Figure 5.

Figure 6: Please select the time and tempeture that can be compared.

Abstract:

P1L18-19: This sentence should be rewritten.

P1L24: Please delete “,”.

P1L24-25: Cold injury was more serious in the CaSPDS-down-regulated seedlings ?

Introduction:

In this section, some important terms in the article were briefly introduced. However, the main experimental methods should be introduced. Please adjust the structure order of the introduction. In addition, some sentences need to be improved:

P2L47: Please add s after response.

P2L67: Please delete  “stress” after  “salt”.

P2L71-72: Please write “in pepper” at the end of the sentence.

P2L75: “has become”? There are many grammatical errors in the article. Please check carefully.

P2L78-79: “is significantly increased”? The tense of this sentence is inconsistent.

Results:

Please write the results behind the materials and methods.

P4L121-123: Please explain the significance of comparison and counterpart.

P5L136-150: Please add a comparison figure of “Expression of CaSPDS with and without cold stress”.

P6L168-169: “under which ……”There is a grammatical error in this sentence.

Discussion:

In the section, some references cited in the discussion may not be the latest achievements.

All results have not been fully discussed.

There is a lot of repetition of the results in the discussion.

P10L308-310: Please rewrite this sentence.

P11L328-329: “but no significant difference in chilling resistance was observed between the WT plants and OE lines.” What is the role of this sentence?

Materials and Methods

The experimental design was well shown and the experiment was carried out carefully. It is advisable to record all the temperature conditions under which the experiment was conducted.

Conclusion:

“CaSPDS is valuable in regulating the cold tolerance of pepper. ” Does this conclusion apply to all varieties of pepper?

Author Response

Please see the attachment:

Response to Reviewer 3 Comments

This manuscript deals with the CaSPDS in pepper plays an important and positive role in response to cold stress. This work provides new sight and opinion into the development of molecular breeding to enhance the cold tolerance of pepper. However, several points need to be improved. Below are comments for the authors to improve their manuscript.

Tables and figures:

Figure 1: Please replace “LOC107847831” with “LOC107847831”.

Thank you very much for your comment. We have revised the Figure 1.

Figure 2: Figure 2B is unclear. Please supplement the analysis results of phylogeny.

Thank you very much for your comment. We have revised the Figure2B, and added the analysis results of phylogeny.

Figure 3: Please replace “CaSPDS” with “CaSPDS”, which is also need to be check in the text, such as Figure 4, Figure 5.

Thank you very much for your comment. We have revised the Figure 3, Figure 4, and Figure 5.

Figure 6: Please select the time and tempeture that can be compared.

 Thank you very much for your comment. This part has been compared with different temperatures, that is CK: normal temperature, 4 ℃: chilling stress, −8 ℃: freezing stress. The results showed that CaSPDS-overexpression Arabidopsis plants were more tolerant to freezing stress.

Abstract:

P1L18-19: This sentence should be rewritten.

Thank you for your good suggestions of our manuscript. We have revised. The details were in the manuscript by marking in red.

P1L24: Please delete “,”.

Thank you for your good suggestions of our manuscript. We have deleted.

P1L24-25: Cold injury was more serious in the CaSPDS-down-regulated seedlings ?

 Thank you very much for your comment. We have replaced CaSPDS-down-regulated with CaSPDS-silenced. The details were in the manuscript by marking in red.

Introduction:

In this section, some important terms in the article were briefly introduced. However, the main experimental methods should be introduced. Please adjust the structure order of the introduction. In addition, some sentences need to be improved:

P2L47: Please add “s” after “response”.

Thank you very much for your comment. We have revised. The details were in the manuscript by marking in red.

P2L67: Please delete  “stress” after  “salt”.

Thank you very much for your comment. We have deleted.

P2L71-72: Please write “in pepper” at the end of the sentence.

Thank you very much for your comment. We have revised. The details were in the manuscript by marking in red.

P2L75: “has become”? There are many grammatical errors in the article. Please check carefully.

Thank you very much for your comment. We have revised. The details were in the manuscript by marking in red.

P2L78-79: “is significantly increased”? The tense of this sentence is inconsistent.

Thank you very much for your comment. We have revised. The details were in the manuscript by marking in red.

Results:

Please write the results behind the materials and methods.

Thank you very much for your comment. Generally, the submission format of IGMS is the results in front of the materials and methods.

P4L121-123: Please explain the significance of comparison and counterpart.

Thank you very much for your comment. First of all, we are very sorry for the ambiguity caused by our non-standard writing. We use the "counterpart" to refer to the SPDS protein sequence of Solanaceous plants. The revised details were in the manuscript by marking in red.

P5L136-150: Please add a comparison figure of “Expression of CaSPDS with and without cold stress”.

Thank you very much for your comment. As shown in Figure 3B, pepper seedlings were subjected to cold stress at 4 ℃, followed by 0, 3, 6, 12, and 24 h. Among them, 0 h is taken as the control check (CK), that is, sampling at room temperature without cold treatment.

P6L168-169: “under which ……”There is a grammatical error in this sentence.

Thank you very much for your comment. We have revised. The details were in the manuscript by marking in red.

Discussion:

In the section, some references cited in the discussion may not be the latest achievements.

Thank you very much for your comment. We have We have added some of the latest references. The details were in the manuscript by marking in red.

All results have not been fully discussed.

Thank you very much for your comment. We have added some content for discussion. The details were in the manuscript by marking in red.

There is a lot of repetition of the results in the discussion.

Thank you very much for your comment. We have deleted the contents of repeated discussions. The details were in the manuscript by marking in red.

P10L308-310: Please rewrite this sentence.

Thank you very much for your comment. We have revised. The details were in the manuscript by marking in red.

P11L328-329: “but no significant difference in chilling resistance was observed between the WT plants and OE lines.” What is the role of this sentence?

Thank you for your good suggestions of our manuscript. We have deleted this sentence. The details were in the manuscript by marking in red.

Materials and Methods

The experimental design was well shown and the experiment was carried out carefully. It is advisable to record all the temperature conditions under which the experiment was conducted.

Thank you very much for your comment. We have recorded the time and temperature of cold treatment for future reference.

Conclusion:

“CaSPDS is valuable in regulating the cold tolerance of pepper. ” Does this conclusion apply to all varieties of pepper?

 Thank you very much for your comment. Previous studies have shown that heterologous overexpression of SPDS gene in apple and Arabidopsis can improve the cold resistance of transgenic plants [1-2]. Based on the results of this experiment and the high conservation of the SPDS gene, we speculate that CaSPDS in our experiment is involved in the regulation of cold response of pepper. However further studies are needed to clarify this.

[1] Kasukabe, Y.; He, L.; Watakabe, Y.; Otani, M.; Shimada, T.; Tachibana, S. Improvement of environmental stress tolerance of sweet potato by introduction of genes for spermidine synthase. Plant Biotechnol. 2006, 23, 75-83 [doi: 10.5511/plantbiotechnology.23.75].

[2] Kasukabe, Y.; He, L.X.; Nada, K.; Misawa, S.; Ihara, I.; Tachiban, S. Overexpression of spermidine synthase enhances tolerance to multiple environmental stresses and up-regulates the expression of various stress-regulated genes in transgenic Arabidopsis thaliana. Plant cell physiol. 2004, 45, 712-722 [doi: 10.1093/pcp/pch083].
